# Influence of Cedar Essential Oil on Physical and Biological Properties of Hemostatic, Antibacterial, and Antioxidant Polyvinyl Alcohol/Cedar Oil/Kaolin Composite Hydrogels

**DOI:** 10.3390/pharmaceutics14122649

**Published:** 2022-11-29

**Authors:** Tamer M. Tamer, Maysa M. Sabet, Zahrah A. H. Alhalili, Ahmed M. Ismail, Mohamed S. Mohy-Eldin, Mohamed A. Hassan

**Affiliations:** 1Polymer Materials Research Department, Advanced Technologies and New Materials Research Institute (ATNMRI), City of Scientific Research and Technological Applications (SRTA-City), New Borg El-Arab City, Alexandria 21934, Egypt; 2Central Laboratory, Faculty of Agriculture, Ain Sham University, Cairo 11241, Egypt; 3Department of Chemistry, Faculty of Sciences and Arts in Sajir, Shaqra University, Dawadmi 11912, Saudi Arabia; 4Basic Science Department-Arab Academy for Science, Technology and Maritime Transport, Aswan Branch, Aswan 81511, Egypt; 5Protein Research Department, Genetic Engineering and Biotechnology Research Institute (GEBRI), City of Scientific Research and Technological Applications (SRTA-City), New Borg El-Arab City, Alexandria 21934, Egypt; 6University Medical Center Göttingen, Georg-August-University, 37073 Göttingen, Germany

**Keywords:** PVA, hydrogel, cedar essential oil, kaolin, fast clotting dressing, wound dressing

## Abstract

Polyvinyl alcohol (PVA) is a safe and biodegradable polymer. Given the unique physical and chemical properties of PVA, we physically cross-linked PVA with kaolin (K) and cedar essential oil (Ced) using the freeze-thawing approach to fabricate PVA/Ced/K sponge hydrogels as hemostatic, antibacterial, and antioxidant wound healing materials. The physicochemical characteristics of PVA/Ced/K hydrogels, including water swelling profiles and gel fractions, were surveyed. Additionally, the functional groups of hydrogels were explored by Fourier transform infrared spectroscopy (FTIR), while their microstructures were studied using scanning electron microscopy (SEM). Furthermore, the thermal features of the hydrogels were probed by thermal gravimetric analysis (TGA) and differential scanning calorimetry (DSC). Evidently, alterations in cedar concentrations resulted in significant variations in size, water uptake profiles, and hydrolytic degradation of the hydrogels. The incorporation of cedar into the PVA/K endowed the hydrogels with significantly improved antibacterial competency against *Bacillus cereus* (*B. cereus*) and *Escherichia coli* (*E. coli*). Moreover, PVA/Ced/K exhibited high scavenging capacities toward ABTS^•+^ and DPPH free radicals. Beyond that, PVA/Ced/K hydrogels demonstrated hemocompatibility and fast blood clotting performance in addition to biocompatibility toward fibroblasts. These findings accentuate the prospective implementation of PVA/Ced/K composite hydrogel as a wound dressing.

## 1. Introduction

The skin’s typical response to injury is a reinstatement process identified as wound healing. Toward this end, hemostasis, inflammation, cellular proliferation, and tissue remodeling are the dominant interconnected stages of the cutaneous healing process [1,2].

Initially, avascular contraction and the blood coagulation episodes immediately follow a skin injury, stopping further bleeding and hampering the invasion of pathogenic microorganisms. Additionally, blood clots function as a scaffold for dermal cells to migrate toward the wound site for wound healing and further tissue remodeling [3,4], as well as a source of growth factors and vital cytokines for this process. However, bleeding has been the major cause of death among both civilians and soldiers for decades. Most deaths in trauma patients arise during the first hour after the injury [5,6], making instant hemorrhage management through blood clotting intervention a critical mechanism. Uncontrolled bleeding also has serious consequences, such as wound inflammation and microbial infections [7], which all stall the healing process.

The hemostatic process includes the following sequential episodes: initiation, which comprises the production of thrombin, followed by amplification to activate and aggregate the platelets and, finally, proliferation, which involves the enhancement of fibrin and the establishment of the platelet clot, all of which are coordinated in a specific order. The majority of therapeutically utilized hemostatic drugs serve an essential function by inspiring the aggregation of platelets and coagulation throughout the amplification and proliferation stages of the bleeding process [8]. Furthermore, some of the most important qualities in a hemostatic agent are its simplicity of application, low price, compatibility with blood and other tissues, and compatibility with cells [9].

Paramount characteristics are required for rapid hemostasis, including quick and strong adhesion to govern blood flow and great mechanical potency to sustain blood pressure, as well as favorable biocompatibility to promote tissue regeneration. Wet and dynamic tissue surfaces, on the other hand, are extremely difficult to bond [10,11].

Given this fact, hemostatic wound dressings have been receiving critical attention to induce blood clotting. Furthermore, they could develop a physical impediment between the injuries and the surrounding environment, thwarting expanded wounds or microbial infections [12]. Several types of wound dressings have been designed with different formulations, such as membranes, electrospun nanofibers, and hydrogel wound dressings [13,14]. However, wound dressings based on hydrogels have particular advantages among other types of dressings due to their capability to soak up the surplus of injury exudates, provide the surface of the injury with cool conditions for pain relief, and sustain the balance of moisture at the wound bed for boosting the migration of dermal cells and their propagation [15,16].

Moreover, given the remarkable adhesive feature of hydrogels, they are predominantly applied as hemostatic dressings in cases suffering from excessive bleeding [17,18,19]. This bestows hydrogels with the capability to efficiently seal and fill wounds with irregular shapes in addition to non-compressible visceral bleeding injuries. In contrast to conventional fibrin-containing hemostatic materials, which function well under typical coagulation conditions [20,21], adhesive hydrogels can promptly produce an adhesion hindrance through tissue interaction to firmly block the hemorrhaging spot, which is primarily dependent on thrombin and fibrinogen in the blood [22].

Injectable hydrogels have the ability to act as hemostatic materials [18], but their lack of sufficient mechanical features constrains their usage in practice. The pain associated with replacing or removing injectable hydrogels is another potential shortcoming for patients, mainly for those who have suffered severe wounds [23]. Consequently, it is beneficial to develop 3D sponges with antibacterial, hemostatic, and antioxidant activities due to their unique structures, which support them with some mechanical stability. It is also vital to highlight that the dimensionality of the structure aids in the adhesion and proliferation of cells brought in to aid in wound healing [9].

Several formulations for hemostatic and antibacterial wound dressings based on hydrogel composites supported by inorganic materials were reported. Haidari et al. introduced multifunctional thermo-responsive hydrogels inspired by ultrasmall silver nanoparticles (size < 3 nm) as antibacterial and anti-inflammatory wound dressings, which frustrated the growth of different virulent bacteria and accelerated the wound healing process of infected wounds [24,25]. Cheng et al. fabricated an antibacterial wound dressing with high hemostatic efficacy using an agar–polyvinyl alcohol hydrogel inspired by tannic acid [26]. Additionally, previous studies presented antimicrobial, antioxidant, and hemostatic PVA and chitosan hydrogels with adhesive and self-healing potency for enhancing the restoration of wounded skin [27,28]. Furthermore, multifunctional hydrogel wound dressings derived from carboxymethyl chitosan and oxidized dextran/sodium alginate substantiated their significant role as hemostatic and antibacterial dressings for promoting wound closure in pathogenic bacteria-infected wounds [29,30].

Porous sponges established on the basis of polyvinyl alcohol (PVA) have demonstrated excellent mechanical properties and outstanding biocompatibility [31]. To avoid the usage of harmful chemical cross-linking materials that could be detrimental for some critical applications, physical cross-linking could be applied for the development of PVA sponges. In this context, PVA-based sponges may be cross-linked through a process of sequential freeze-thawing, with crystalline clusters serving as the network point [32,33]. Integrating a hemostatic agent, such as kaolin, that has the ability to stimulate the accumulation of both blood cells and platelets, in addition to bioactive materials (cedar oil) to bestow the hydrogels with both antibacterial properties and scavenging competency for free radicals to thwart inflammation in the injury site, is crucial.

Kaolin has been acknowledged as a substantial hemostatic that can significantly boost blood coagulation [34]. China clay, or kaolin, is a type of clay that is mostly made up of kaolinite and aluminum silicate [34]. Kaolin was widely employed as a functional compound in hemostasis owing to the fact that the negative charges on its surface could significantly stimulate the coagulation of blood. Specifically, kaolin has been shown to induce factor XII and platelets, two more players in the blood clotting process [35].

Cedar oil is an essential oil extracted from numerous types of conifers and possesses some pesticidal properties [36]. As a food additive and preservative, cedarwood oil is a blend of organic chemicals classified as safe compounds by the FDA [37].

We assumed that the synergistic influence of the combination of PVA, cedar oil, and kaolin may adequately hinder hemorrhage and bacterial infections during the normal cascade of wound recovery, in conjunction with PVA’s ability to heal the wound, thereby promoting the action of the designed sponge. As a result, we developed groups of porous PVA/cedar/kaolin composite sponges that could be applied as hemostatic and antibacterial wound dressings while also modulating reactive oxygen species (ROS) through free radical scavenging activity. The physicochemical and biological traits of sponges were also analyzed to assess their implementation as favorable wound dressings.

## 2. Materials and Methods

### 2.1. Materials

Here, PVA (Mw = 72 kDa) was obtained from ACROS Organics^TM^, Carlsbad, CA, USA. Oil of cedar and pure ethanol were supplied by Sinopharm Chemical Reagent Co., Ltd. (Beijing, China). Kaolin (hydrated aluminum silicate), Folin–Ciocalteu, and gallic acid, were supplied by Sigma-Aldrich Co., Darmstadt, Germany. The DPPH and ABTS were acquired from Sigma-Aldrich Co., Ltd. in St. Louis, MI, USA.

#### Bacterial Strains

*Bacillus cereus* (*B. cereus*) and *Escherichia coli* (*E. coli*) were kindly provided by the Genetic Engineering and Biotechnology Research Institute (GEBRI), City of Scientific Research and Technological Applications (SRTA-City), New Borg El-Arab City, Alexandria, Egypt. These strains were used to investigate the antibacterial performance of wound dressings. To refresh the bacterial strains before being applied, the bacteria were grown overnight in LB medium at 37 °C under shaking conditions at 150 rpm.

### 2.2. Methods

#### 2.2.1. Preparation of Composite Sponge’s Hydrogel

The PVA/cedar/kaolin composite sponges were developed using a freezing-thawing cycle technique, as described in our previously published methods [38,39]. Different amounts of cedar oil (0.1, 0.25, and 0.5 mL) and kaolin (0.25 g) were added to 50 mL of PVA (5%) solution. The mixture was thoroughly mixed, followed by sonication for 1 h before being cast in plastic Petri dishes.

Five cycles of freezing (at −20 °C) and thawing (at +25 °C) were conducted for 18 h and 6 h, respectively. With blank PVA hydrogel, three different formulated sponges with varying amounts of cedar (0.1, 0.25, and 0.5 mL) were prepared and termed PVA/Ced0.1/K, PVA/Ced0.0.25/K, and PVA/Ced0.5/K, respectively. The sponges were flash-frozen in liquid nitrogen for 10 min before being lyophilized for further examinations.

#### 2.2.2. Characterization of the Sponges

For FTIR measurements, 6 mg of sponge was completely mixed with potassium bromide and then analyzed using a Shimadzu 8400S, Kyoto, Japan, ranging from 400 to 4000 cm^−1^.

To inspect the morphological changes in the sponges, a scanning electron microscope (SEM, Joel JSM 6360LA, Tokyo, Japan) was used.

A TGA, Shimadzu 50/50H, Kyoto, Japan, was used for thermal characterization across a temperature range from 20 to 600 °C.

For gel fraction evaluation, a defined amount of sponge was dried for 24 h at 50 °C in a vacuum oven prior to being weighed. The sponges were subsequently immersed in distilled water for 24 h until reaching the equilibrium swelling level to remove the soluble PVA. Afterwards, the specimens were dried at 50 °C and then weighed.

The swelling capabilities of sponges were assessed by determining their weights after submerging them in water for a period of time. The swelling ratio was performed and calculated, as demonstrated in our previous study [38].

To assess the hydrolytic decomposition of the designed sponges, dry samples of sponges were weighed, and soaked in PBS, at 37 °C. The samples were then removed and dried before being weighed. Accordingly, the weight loss was calculated. All experiments were completed with five independent replications.

#### 2.2.3. Bioactive Evaluations of the Sponges

The antibacterial capability of the sponges was estimated following the determination of the optical densities of bacterial strains to further evaluate the bacterial growth inhibition. First, overnight *E. coli* and *B. cereus* cultures were diluted using LB medium to adapt the turbidity of cultures at 625 nm following the McFarland 0.5 standard [40,41]. Then, 100 µL of the bacterial suspensions were transferred into 10 mL of LB medium, which contained 50 mg of the examined dressings, followed by incubations of the bacterial tubes overnight at 200 rpm and 37 °C. The bacterial cultures were then measured using a spectrophotometer at 600 nm.

To determine the phenolic content in the prepared hydrogels, 50 mg of film was separately immersed in 5 mL of ethanol in order to complete the release of the hydrogel content represented by the cedar oil. After that, 0.5 mL of ethanolic extract was then mixed with 2.0 mL of Folin–Ciocalteu reagent before being mixed with 2 mL of Na_2_CO_3_ (7.5%, *w*/*v*). The mixture was then stirred for 5 min at 50 °C, followed by gauging the density of color at 760 nm using a spectrophotometer. The measurements were repeated and calculated according to standard gallic acid.

For the ABTS^•+^ radical scavenging evaluation, 0.1 mL of the extracted solution prepared in the previous test (total phenolic content test) was blended with the solution of ABTS^•+^ (2.0 mL) [42]. The ABTS^•+^ reaction was conducted for 5 times and gauged at 730 nm. The absorbance was evaluated at different time points.

Adapting the 2,2-diphenyl-1-picrylhydrazyl (DPPH) method [43], the antioxidant performance of the sponge extract was evaluated. Briefly, 2.0 mL of each ethanolic extract was blended with 2.0 mL of the DPPH reagent and kept for 20 min at room temperature in the dark. Following this, the reaction was then gauged using a spectrophotometer at 517 nm.

To examine the hemocompatibility of the hydrogels, the hemolysis experiments were accomplished as reported earlier with slight amendments [44]. For this evaluation, the hemolysis of blood in the presence of sponges was evaluated in comparison to the positive and negative controls of blood treated with PBS and water, respectively.

Additionally, a gravimetric approach was utilized to determine the quantity of thrombus on contact with the designed sponges [39]. Blood samples were produced as illustrated earlier [38]. The PVA/Ced/K and PVA sponges were soaked in PBS for 48 h at 37 °C. Subsequently, the PBS was discarded, and the blood was placed over the inspected specimens. Simultaneously, a positive control was established by adding the equivalent volume of blood to an empty Petri dish. Next, 20 µL of a 10 M calcium chloride solution was applied to the sponges to instigate blood clotting. After 45 min, the reactions were ceased by the addition of 5 mL of water. The clots were then anchored by adding 5 mL of a formaldehyde solution (36% formaldehyde), dried, and weighed. Examinations of thrombogenicity were repeated five times.

To quantify the toxicity of the designed sponges toward mouse fibroblast cells (NIH-3T3), MTT [3-(4,5-Dimethythiazol-2-yl)-2,5-Diphenyltetrazolium Bromide] assessment was conducted following the previous protocol with minor amendments [45,46]. The NIH-3T3 cells were seeded at 3 × 10^4^ cells/well in a 96-well plate comprising Dulbecco’s modified Eagle’s medium (DMEM) with 10% fetal bovine serum, prior to being incubated for 24 h in a CO_2_ incubator (85% humidity and 37 °C). Meanwhile, 25 mg of each examined hydrogel was immersed in 70% ethanol before being exposed to UV for 2 h. Thereafter, the hydrogel was placed into a 24-well plate comprising 1 mL of DMEM at 37 °C for 24 h to obtain a leachate from each hydrogel. To treat the fibroblast cells with the supernatants extracted from the hydrogels, the medium in a 96-well plate was aspirated, and then each well was provided with 100 µL of the sponge’s suspension, whereas the untreated cells were furnished with 100 µL of DMEM medium. After incubation of the plate for 24 h, the media were aspirated and then washed with PBS before being supplied with 20 µL of MTT solution (5 mg/mL) dissolved in serum-free medium for each cell. After incubation for 3 h in the CO_2_ incubator, the MTT solution was discarded, and each well was then supplied with 200 µL of dimethylsulfoxide (DMSO) before being measured at 570 nm. The cytotoxicity evaluations were performed in six replicates, and the percentage of viable cells was quantified using the following equation:
Cell viability (%) = (Am/Ac) × 100(1)
where (Am) indicates the absorbance of fibroblasts treated with a sponge, while (Ac) refers to the absorbance of control fibroblasts.

#### 2.2.4. Statistical Analysis

To examine the statistical significance of the full dataset, GraphPad Prism (Version 5, GraphPad Software Inc., San Diego, CA, USA) was utilized. Thus, one-way and two-way analyses of variance (ANOVA) in conjunction with Tukey’s analysis for multiple comparisons were utilized. All data are reported as mean ± SD, and they were considered significant at *p* ≤ 0.05.

## 3. Results and Discussion

Previously, we showed the hemostatic feature of PVA/Kaolin sponges and their antibacterial properties, which were enhanced by penicillin–streptomycin [39]. Later, novel PVA/marjoram/kaolin sponges were developed, in which the antibacterial and antioxidant activities of hydrogels were boosted by marjoram essential oil [38]. In this investigation, we devised, for the first time, novel composite sponges of PVA/Kaolin bolstered by cedar oil with competitive biological activity. Specifically, we formulated novel cross-linked sponges by freezing and thawing PVA inspired by cedar extract and kaolin. In addition, because PVA lacks antibacterial, antioxidant, and hemostatic properties, cedar oil and kaolin were added to the developed sponges.

### 3.1. Characterization

Figure 1 illustrates the FTIR spectra of PVA sponge hydrogel in addition to the PVA/cedar/kaolin composite sponges. The occurrence of stretching vibration bands at 3200–3400 cm^−1^ is related to –OH in PVA chains, which endows the PVA sponge with hydrophilic properties [47,48]. Additionally, asymmetrical and symmetrical C-H stretching vibrations related to methyl groups were detectable at 2930 cm^−1^. In addition, the band at 2841 cm^−1^ correlates to the -CH_2_ vibration band, while the remaining acetyl carbonyl groups were observed at 1712 cm^−1^. The asymmetrical and symmetrical CH bending vibrations imputed to the methyl group’s band were recorded at 1450 cm^−1^ [48].

In addition, the large peak which emerged at 1120 cm^−1^ is the most significant indication of the PVA structure [39], whereas a peak at 1085 cm^−1^ was assigned to C–O–C. The amalgamation of cedar into PVA resulted in the formation of a new peak at 1650 cm^−1^, which combined with the bands of acetyl carbonyl groups in the PVA. Obviously, this ring became stronger with the incorporation of higher concentrations of cedar. In contrast, the amalgamation of kaolin with PVA/cedar hydrogels gave rise to the development of Al-OH vibration-related peaks at 920 to 940 cm^−1^. Furthermore, the peaks at 530 and 789 cm^−1^ are ascribed to the vibration band of the Si-O-Al bond.

The SEM technique was utilized to examine the microstructures of developed sponges. Pure PVA sponges have a morphological surface with fewer pores than PVA sponge mixtures. As shown in Figure 2, the PVA hydrogel composites combined with varying proportions of cedar alongside kaolin exhibited three-dimensional structures linked with varying pore diameters in asymmetrical patterns. In addition, as depicted in Figure 3, cross-sectional images for PVA/Ced/K sponges revealed asymmetric tidy structures with conspicuous 3D linked networks. In addition, the composite sponges demonstrated a porous sponge structure with obvious lamellar structures, which is comparable to those previously applied PVA hydrogels with striking wound healing efficacy [49]. On the other hand, hemostatic wound dressings necessitate a good connection to the injuries for absorbing the overflow exudates alongside their interactions with blood to promote the hemostatic effect [49,50].

Figure 4 illustrates the DSC values for PVA/Ced/K composite hydrogels in comparison to the PVA hydrogel. According to the DSC curves, wide endothermic peaks in the 70–80 °C temperature range are assigned to the evaporation of water molecules confined inside the hydrogel molecules. These results are entirely consistent with those of the prior research. The exothermic peaks which materialized between 70 and 140 °C may be elucidated by relaxation related to crystalline areas in the sponge [51,52]. The endothermic peaks for PVA at 217 °C, PVA/Ced0.1/K at 119 °C, PVA/Ced0.25/K at 223 °C, and PVA/Ced0.5/K at 220 °C implied melting and crystal structural deformation (Tm), which are in line with earlier observations. In addition, the changes in the Tm value of PVA/Ced/K composites toward higher temperatures are indicative of the effect of the addition of cedar oil on the cross-linking density and the crystallinity between PVA chains [53]. The subsequent exothermic peak is connected with the heat degradation of PVA and the loss of water molecules along the backbone of the polymer. According to DSC analyses, the interaction between volatile decomposition products, including water vapor, carbon monoxide, and carbon dioxide throughout the disintegration progression, cedar oil, and kaolin particles may lead to the falling down of peaks. This explanation is in line with the mechanism for chain-stripping [54].

The thermal gravimetric analysis of PVA and PVA/Ced/K sponge hydrogels is delineated in Figure 5. From room temperature to 200 °C, 10% of the weights of PVA and PVA/Ced/K sponges were measured when the first weight loss began. This is probably due to the elevation of water captured by hydrophilic groups (i.e., hydroxyl) in the chains of the polymer. The entire hydrogel lost weight between 220 and 320 °C due to the elimination of -OH groups and the development of polyene complexes. These findings align with those of the prior research [48]. The PVA hydrogel had the greatest weight reduction as compared to the PVA/Ced/K composites. The PVA film lost 75.9% of its original weight between 226–333.6 °C and 219–371 °C, whereas PVA/Ced0.1/K weight decreased by 72.76% between 219–371 °C. In addition, the weight of PVA/Ced0.25/K degraded by approximately 61.93% between 242 and 397 °C, whereas the decomposition ratio of PVA-K0.5 was 59.2% between 242 and 385 °C. It was evident that the weight diminution rate decreased when the cedar oil concentration in the composites increased. At 600 °C, the third phase of deterioration was seen, owing to the decomposition of the produced polyenes. In this stage, the rise in residual weights from 1.5% for pure PVA to 5.2% for PVA/Ced0.1/K, 8.1% for PVA/Ced0.25/K, and 14.7% for PVA/Ced0.5/K is ascribed to the stability of inorganic cedar oil and kaolin residues. Altogether, PVA/Ced0.5/K exhibited the highest thermal stability.

### 3.2. Gel Fraction, Swelling Behaviour, and In Vitro Degradation

The influence of different concentrations of cedar oil on sponges was determined. According to the findings, increasing the amount of kaolin and cedar incorporated into PVA led to a rise in gel fractions. The gel fraction for PVA was 87.58 ± 4.38%, while the gel fractions for PVA/Ced0.1/K, PVA/Ced0.25/K, and PVA/Ced0.5/K were 81.33 ± 4.07%, 77.33 ± 3.87%, and 75.48 ± 3.77%, respectively. This is most likely due to the distorting action of cedar oil and kaolin on the PVA crystal structure. This indicates that in the absence of cedar oil and kaolin [55], PVA was almost entirely cross-linked, but kaolin and cedar oil reduced cross-linking, augmenting the swelling properties of PVA/Ced/K sponges. These properties suggest that when the prepared sponges are applied, blood and wound exudates can be absorbed quickly. Additionally, a drop in gel fraction is correlated with a lessening in flexibility and gel strength. These findings are supported by prior works [56,57].

Sponges with three-dimensional structures, in addition to their functional groups related to hydrophilic properties, are well recognized for improving the swelling capabilities of the sponges [58]. The swelling performance of the PVA/Ced/K hydrogels in vitro was studied, as shown in Figure 6. The swelling ratios showed substantial decreases for PVA/Ced0.1/K and PVA/Ced0.25/K compared to the PVA sponge. Conversely, the PVA/Ced0.5/K sponge revealed a non-significant difference with regards to the PVA sponge. Overall, these findings support the prospective treatment of wounds with PVA/Ced0.5/K sponges for accelerating wound repair.

The degradation of PVA/Ced/K sponges was examined in vitro by immersing them in PBS at 37 °C for predefined durations. Following incubation for 72 h, the tested sponge hydrogels demonstrated measurable weight losses of 32.35 ± 1.62%, 22.26 ± 1.11%, and 24.95 ± 1.25% for PVA/Ced0.1/K, PVA/Ced0.25/K, and PVA/Ced0.5/K, respectively, as displayed in Figure 7. In comparison, the PVA group lost 28.08 ± 1.40% of their body weight. These findings reveal that the addition of cedar oil has a considerable impact on the deterioration rate of sponges. In this investigation, it is believed that the hydrolytic degradation in vitro may affect the drug release signified in this work by cedar oil, which agrees with previous findings [59,60].

### 3.3. Bioactivity Evaluations of the Hydrogels

The antibacterial competency of the wound dressing is essential in preventing the growth of prevalent pathogenic bacteria throughout the wound healing process, which could impede the regeneration of tissues and may lead to vitiation of skin tissues [61,62]. To that purpose, we inspired the sponges with cedar oil, a naturally antibacterial substance that may be used instead of common antibiotics to inhibit the growth of antibiotic-resistant microorganisms [63]. The antibacterial efficacy of PVA/Ced/K sponges was surveyed against *B. cereus* and *E. coli*. The growth turbidity technique was used to quantify the rate of bacterial growth inhibition, as exhibited in Figure 8. There is a substantial positive link between the antibacterial ability of PVA/Ced/K sponges and an increase in cedar ratio. The pure PVA sponges, in particular, showed no effect against the tested microorganisms. Importantly, adding cedar to PVA/kaolin boosted the antibacterial effectiveness of PVA/Ced0.1/K against *B. cereus* by 35.62%, PVA/Ced0.25/K by 55.14%, and PVA/Ced0.5/K by 84.62%.

On the other hand, PVA/Ced0.1/K, PVA/Ced0.25/K, and PVA/Ced0.5/K sponges also inhibited *E. coli* development by 57.91%, 84.70%, and 90.18%, respectively. The differences in antibacterial ability could be attributed to the variation in cell wall structures between both indicator bacteria.

A surplus of ROS is a deleterious bioburden concerning wound repair, instigating oxidative stress as a consequence of the phagocytosis mechanism [61,64]. This may trigger lipid peroxidation, inactivation of essential enzymes, and DNA damage [65,66], impairing wound healing and neighboring tissues [59]. Thus, growing wound dressings with antioxidant activity is vital for governing ROS during wound repair [67,68,69].

The presence of numerous phenolic compounds in essential oils is an advantageous attribute. These chemicals exclusively endow them with crucial biological functions, including antioxidant properties, to scavenge ROS. In this context, cedar oil has been shown to contain phenolic acids and terpenoids [36]. The total phenolic contents of the tested sponges were evaluated after soaking them in ethanol to investigate the efficacy of PVA/Ced/K sponges in releasing cedar oil into the medium characterized by the phenolic agents. As a result, the sponges’ structural integrity was exploited, and the matching phenolic combinations were emancipated. Figure 9A shows that no phenolic chemicals were found in the PVA sponge, which was set out as a negative control. On the other hand, there is a clear pattern of rising phenolic contents with increasing cedar oil concentrations. 

Figure 9B depicts the time-dependent decolorization of the ABTS^•+^ cationic radical by PVA and PVA/Ced/K sponge ethanol extracts. The pure PVA hydrogel displayed mild scavenging activity with regard to the ABTS^•+^ radical, which may derive from the hydroxyl groups in the backbone of the PVA sponge. Nevertheless, adding cedar oil to sponges markedly increased ABTS^•+^ radical scavenging activity. These findings are in agreement with those reported in the preceding section by measuring total phenolic content. In particular, the phenolic compounds in cedar oil bestow an electron on ABTS^•+^, which engenders the loss of its color and transforms it into a neutral state [70,71,72].

To further evidence the antioxidant potency of PVA/Ced/K groups, an in vitro design system was used to quantify the examined materials’ capacity to abolish free radicals using the DPPH assay. This assay’s methodology is based on the scavenging of the DPPH by converting the DPPH into diphenyl-picrylhydrazine as a result of absorbing an electron from antioxidant chemicals [73]. Figure 9C depicts the DPPH dye scavenging activity of PVA/Ced/K sponges. The findings are comparable to those obtained from the ABTS^•+^ test. Furthermore, the hydroxyl groups in the pure PVA sponges resulted in a weak DPPH dye scavenging ratio. Simultaneously, there are favorable correlations between DPPH scavenging ratios and a rise in cedar oil concentration.

One of the basic qualities necessary in wound dressings is blood compatibility [74]. To study the possibility of inducing hemolysis in RBCs, the hemocompatibility of PVA and PVA/Ced/K hydrogels was assessed. Figure 10A delineates the hemolytic percentages of the examined sponges. There were no significant variations in hemolysis between PVA supported by varied amounts of cedar oil. The statistical analysis, however, revealed no substantial differences in PVA when compared to the PVA/Ced/K hydrogels. Indeed, the PVA/Ced/K sponges showed hemolysis of less than 2%, which is regarded as safe by the American Society for Testing and Materials (ASTM).

As demonstrated in Figure 10B, the thrombogenicity of PVA and PVA/Ced/K hydrogels was investigated. Because of the hydrophilic nature of PVA, PVA sponges have a lesser tendency for thrombus development than blood control. By contrast, the addition of kaolin and cedar oil to PVA hydrogels resulted in a substantial rise in thrombus formation, which stems from the blood clotting activity of kaolin.

The cellular compatibility assay is decisive for exploring how dermal cells, such as fibroblasts recruited for the restoration of skin tissues, react to wound dressings, particularly toward the sponges developed in this study. It is recognized that fibroblasts play a vital role throughout wound repair in developing connective tissues, which further leads to the granulation and regeneration of skin tissues [75,76,77]. Thus, we applied the MTT evaluation to examine cytotoxicity in fibroblast cells to assess their interactions with the designed sponges. The cytotoxicity findings revealed no significant difference among the entire tested sponges in comparison to untreated fibroblasts, as depicted in Figure 10C. Specifically, the viability of fibroblasts after treatment with sponges was greater than 95%. Accordingly, the potential implementation of the devised PVA/Ced/K sponges in wound recovery could be inferred from these results. Altogether, these findings are encouraging for future in vivo investigations of PVA/Ced/K sponges.

## 4. Conclusions

In conclusion, innovative sponges based on PVA and enhanced with cedar oil and kaolin were developed to preclude hemorrhage and bacterial infections. The PVA/Ced/K sponges demonstrated a distinct porous structure with prominent lamellar architectures. The incorporation of cedar and kaolin into PVA increased the pore size of the fabricated sponges. Furthermore, they displayed high water absorption, indicating their ability to control the bleeding promptly. The PVA/Ced/K sponges exhibited antioxidant capacity in terms of free radical scavenging as well as antibacterial activity against pathogenic microorganisms. Furthermore, the produced sponges’ thrombogenicity and hemocompatibility were validated. As a consequence, the results clearly show that the PVA/Ced/K sponges should be considered as a hemostatic, antibacterial, and antioxidant wound dressing in the future. Collectively, future in vivo studies to assess the extent to which PVA/Ced/K could promote cutaneous wound repair in conditions of bleeding and microbial infections are warranted.

## Figures and Tables

**Figure 1 pharmaceutics-14-02649-f001:**
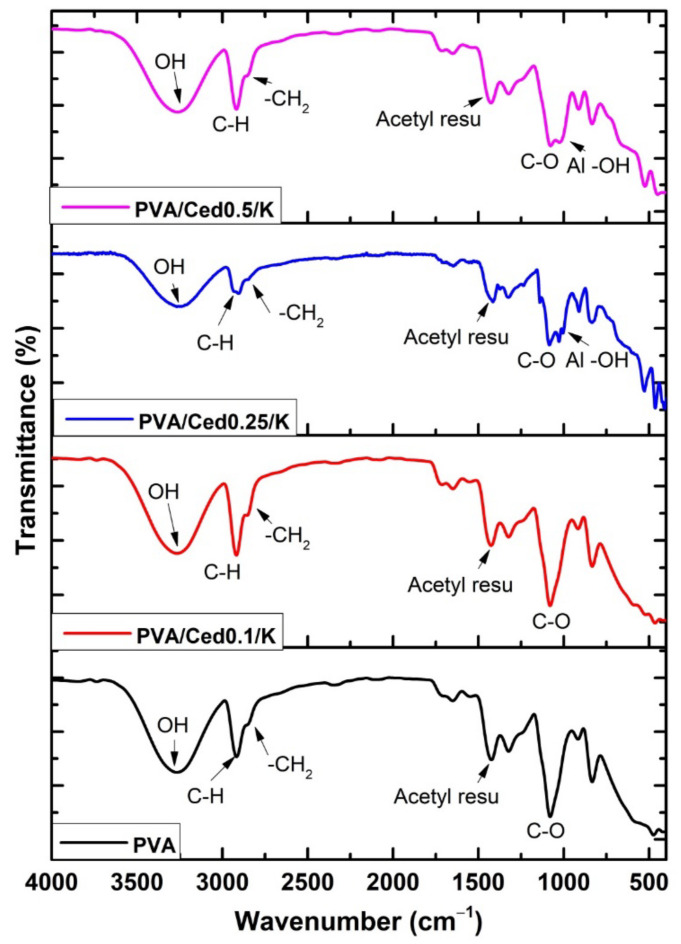
FTIR spectra of different PVA/Ced/K sponges.

**Figure 2 pharmaceutics-14-02649-f002:**
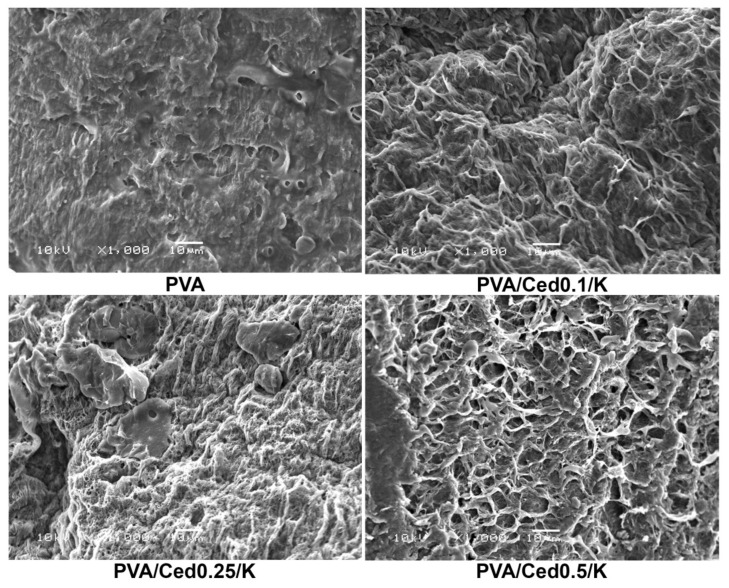
SEM images reveal the surface morphologies of composite sponges.

**Figure 3 pharmaceutics-14-02649-f003:**
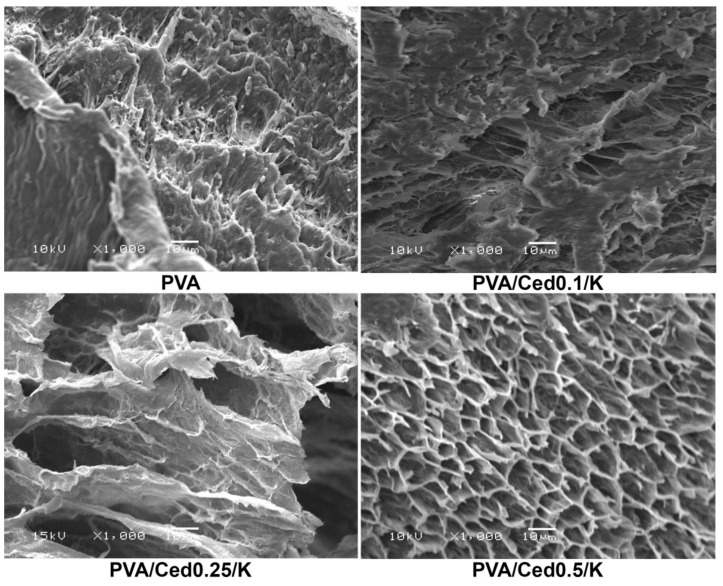
SEM images of cross-sectional areas of composite sponges at a magnification of 1000×.

**Figure 4 pharmaceutics-14-02649-f004:**
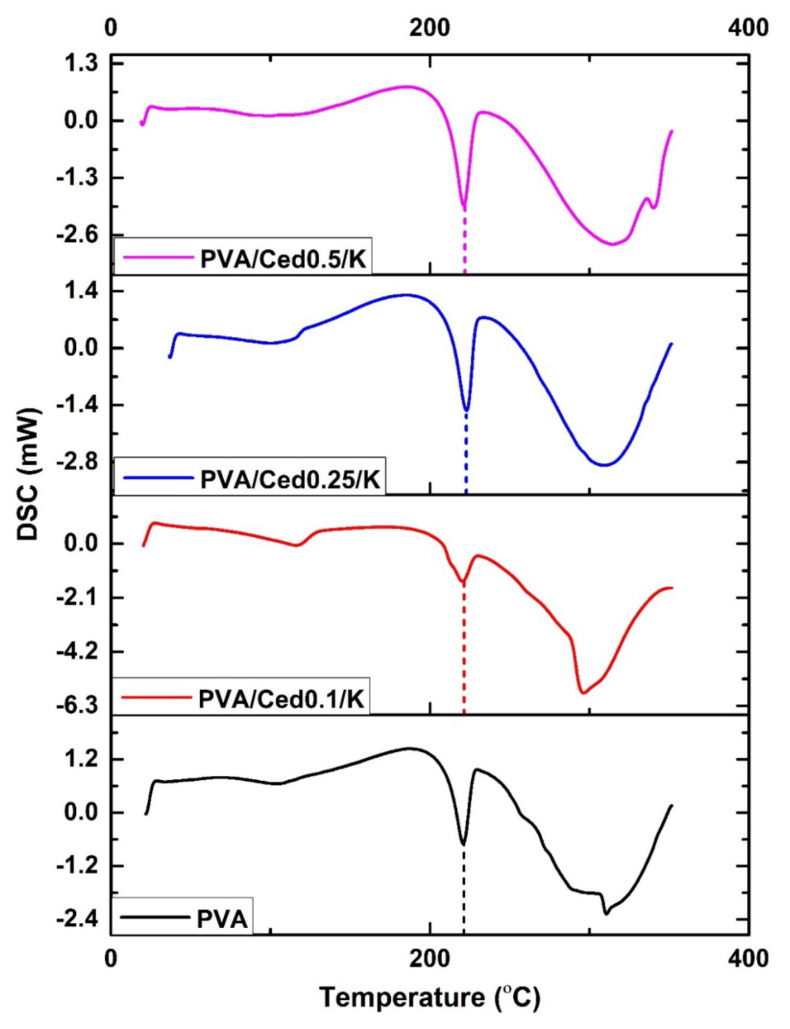
DSC analyses of PVA/Ced/K composite sponges.

**Figure 5 pharmaceutics-14-02649-f005:**
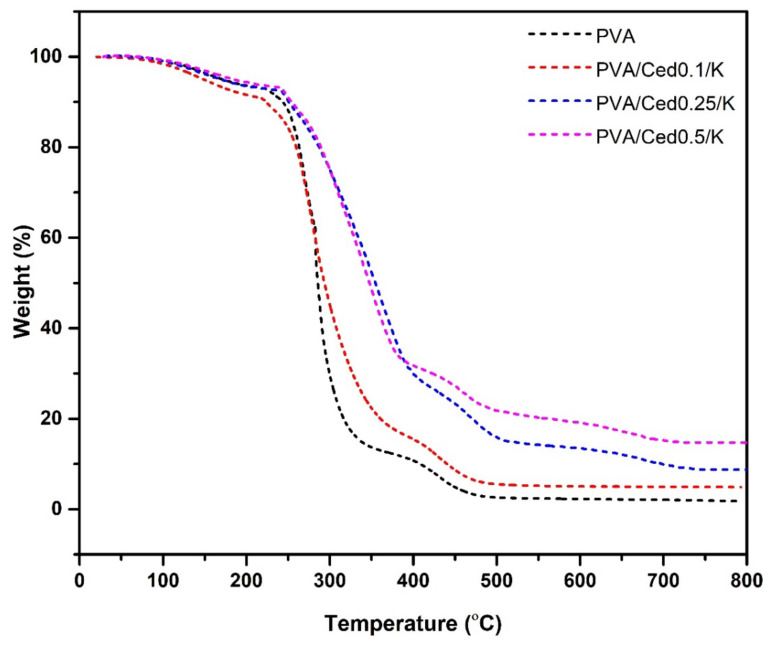
TGA analyses of PVA/Ced/K composite sponges.

**Figure 6 pharmaceutics-14-02649-f006:**
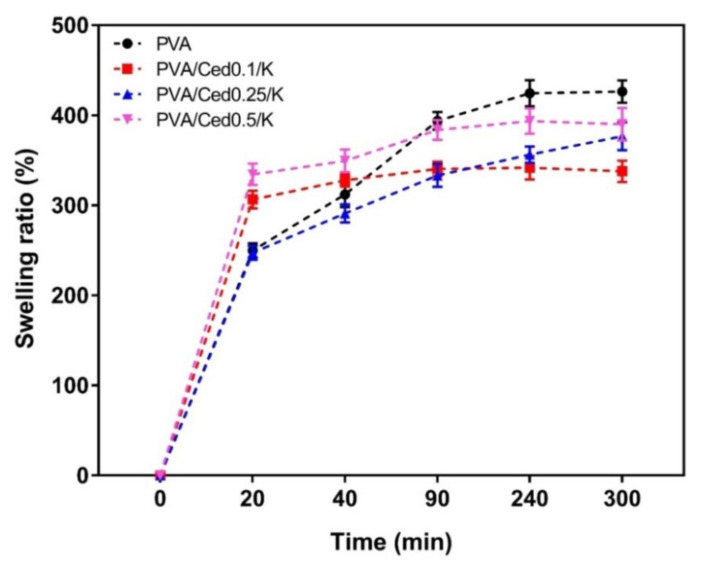
Dynamic water swelling profiles of PVA/Ced/K composite sponges. Results are shown as mean ± SD.

**Figure 7 pharmaceutics-14-02649-f007:**
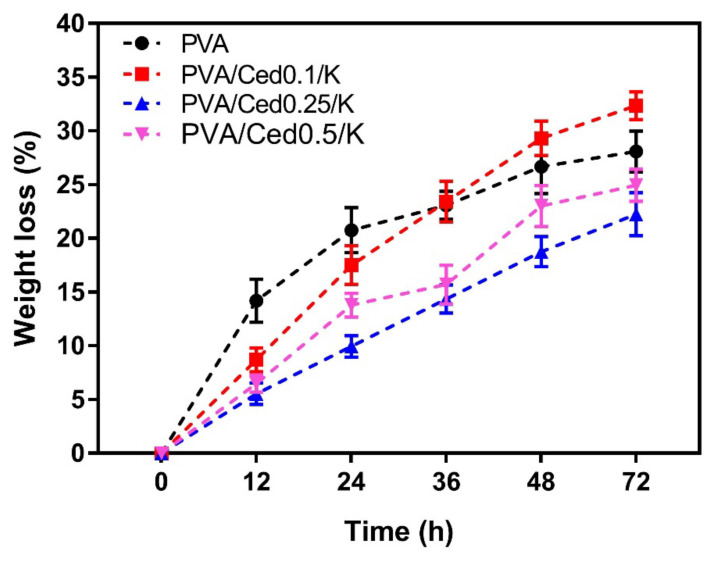
In vitro hydrodegradation of PVA/Ced/K composite sponges. Results are depicted as mean ± SD.

**Figure 8 pharmaceutics-14-02649-f008:**
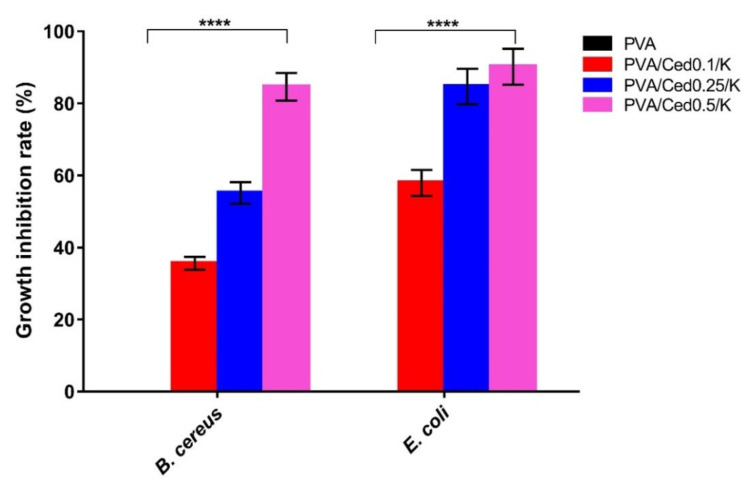
Antibacterial evaluations of PVA/Ced/K composite sponges against *B. cereus* and *E. coli* compared to the PVA sponge. Results are presented as mean ± SD (**** *p* < 0.0001).

**Figure 9 pharmaceutics-14-02649-f009:**
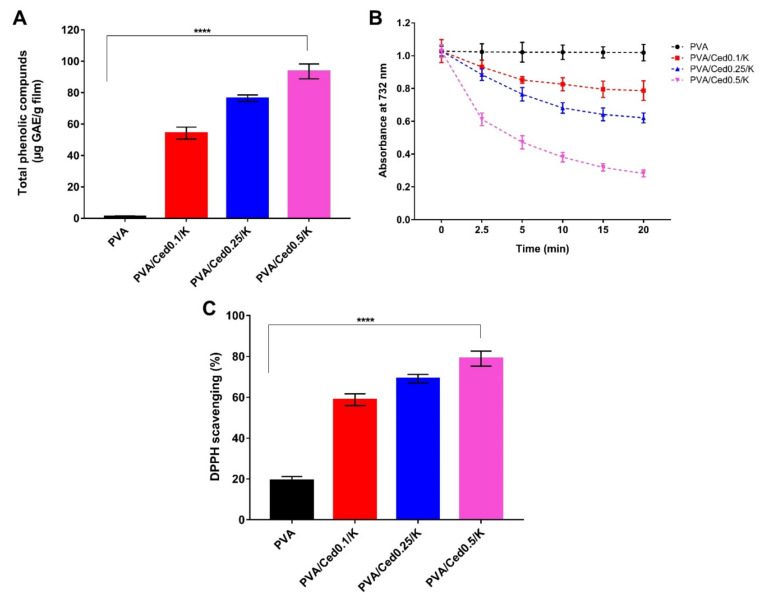
(**A**) Total phenolic compounds released from PVA/Ced/K sponges, (**B**) time-dependent decolourization of ABTS^•+^ dye by PVA/Ced/K sponges, and (**C**) Scavenging properties of PVA/Ced/K sponges against DPPH free radicals. Results are shown as mean ± SD (**** *p* < 0.0001).

**Figure 10 pharmaceutics-14-02649-f010:**
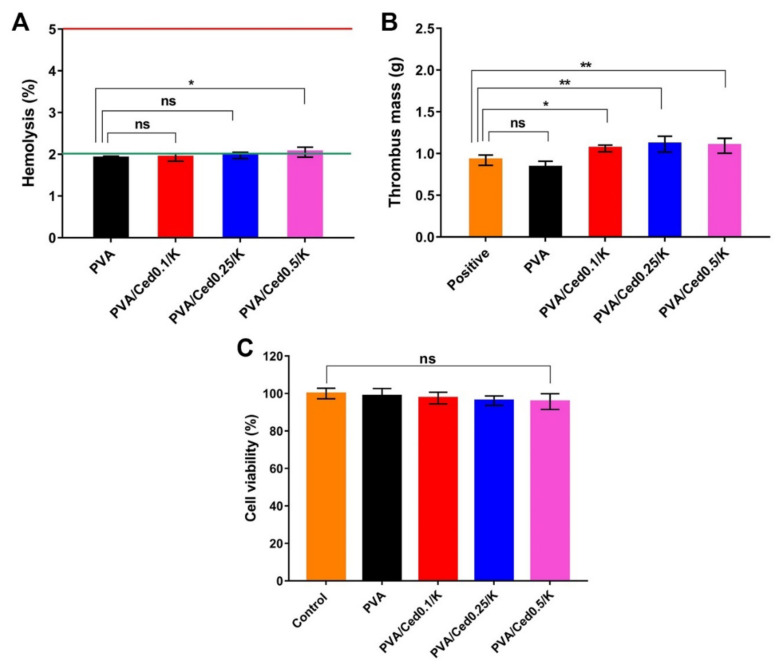
(**A**) Hemocompatibility, (**B**) thrombogenicity, and (**C**) cytotoxicity evaluations for PVA and PVA/Ced/K sponges. Results are expressed as mean ± SD (** *p* < 0.01, * *p* < 0.05, and ns represents a non-significant difference).

## Data Availability

The datasets generated during the current study are available from the corresponding authors upon request.

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
