# Peer review of "Influence of Cedar Essential Oil on Physical and Biological Properties of Hemostatic, Antibacterial, and Antioxidant Polyvinyl Alcohol/Cedar Oil/Kaolin Composite Hydrogels"

_pharmaceutics, 2022, doi:10.3390/pharmaceutics14122649_

Round 1

Reviewer 1 Report

Comments to the authors attached 

Author Response

Many thanks for your valuable comments that guide us in improving the manuscript. We will explain it and answer your questions; please see our response in the attached file.

Many thanks again 

Best regards 

Reviewer 2 Report

The work entitled “Influence of Cedar Essential Oil on Physical and Biological Properties of Hemostatic Antibacterial, and Antioxidant Polyvinyl alcohol/Cedar Oil/Kaolin Composite Hydrogels” reports on the use of Kaolin as a crosslinking agent for the production of PVA hydrogels by freeze-thawing and the incorporation of the cedar essential oil to improve the scaffolding systems hemostatic, antibacterial and antioxidant properties. Data was very promising confirming these biological profiles of the scaffold. The article is of great interest, and it is scientifically sound, however there are some things that require the attention of the authors prior to publication:

-          There is information missing regarding the ratio between polymer, oil and kaolin. It does not say as well how long did the three components were left to blend.

-          Section 2.2 is lacking several details of the parameters employed for examining the scaffolds engineered

-          Section 2.2.3 is also incomplete. Even though the authors report several times to other works, they should provide more detail for the comprehension of their studies. If another researcher wished to replicate their experiments, they would be incapable considering there is much information missing.

-          Why selecting those two bacteria for antimicrobial testing?

-          Cell viability testing is missing from this study. In order to confirm the potentiality of this scaffold to work in wound healing application, cell studies are required (at least with fibroblasts and keratinocytes).

-          Also, the authors made a very significant remark in the abstract regarding the anti-inflammatory properties of the scaffold and no data was provided on that. They should include their findings in that sense or remove that statement.

Author Response

(The authors gave the same response as above.)

Round 2

Reviewer 1 Report

The author has done a great job addressing the queries raised in our first round of review, the paper flows and reads better with improved data presentation. However before we approve for acceptance the author is required to add more references in the introduction and re-check grammatical errors. https://doi.org/10.1016/j.actbio.2021.04.007 

ACS Appl. Mater. Interfaces 2020, 12, 37, 41011–41025 

Author Response

Many thanks for your valuable comments and which help us to improve our manuscript 

Best regards 

Reviewer 2 Report

The authors have implemented all the recommended alterations and the manuscript is now ready for publication.

Author Response

(The authors gave the same response as above.)
